# Venous Thromboembolism in Patients with Neuroendocrine Neoplasms: A Systematic Review of Incidence, Types, and Clinical Outcomes

**DOI:** 10.3390/cancers17020212

**Published:** 2025-01-10

**Authors:** Sara Massironi, Lorenzo Gervaso, Fabrizio Fanizzi, Paoletta Preatoni, Giuseppe Dell’Anna, Nicola Fazio, Silvio Danese

**Affiliations:** 1Faculty of Medicine and Surgery Via Olgettina, Vita e Salute San Raffaele University, 56, 20132 Milan, Italy; fanizzi.fabrizio@hsr.it (F.F.); sdanese@hotmail.com (S.D.); 2Gastroenterology Unit, Istituti Ospedalieri Bergamaschi, 24046 Bergamo, Italy; 3Division of Gastrointestinal Medical Oncology and Neuroendocrine Tumors, European Institute of Oncology (IEO) IRCCS, Via Ripamonti, 435, 20141 Milan, Italy; lorenzo.gervaso@ieo.it (L.G.); nicola.fazio@ieo.it (N.F.); 4Gastroenterology and Endoscopy, IRCCS Ospedale San Raffaele, Via Olgettina, 56, 20132 Milan, Italy; preatoni.paoletta@hsr.it (P.P.); dellanna.giuseppe@hsr.it (G.D.)

**Keywords:** neuroendocrine neoplasms (NENs), thromboembolism, venous thromboembolism (VTE), portal vein thrombosis (PVT), pulmonary embolism (PE), cancer-associated thrombosis

## Abstract

Neuroendocrine neoplasms (NENs) are a heterogeneous group of tumors with distinct biological behaviors, ranging from indolent well-differentiated forms to highly aggressive poorly differentiated variants. Venous thromboembolism (VTE) represents a significant complication in patients with NENs, contributing to morbidity and mortality. This systematic review synthesizes data on the incidence, types, and clinical outcomes of VTEs in this population. Reported VTE prevalence ranged from 7.5% to 33%, with pancreatic NENs demonstrating the highest thrombotic burden, particularly in poorly differentiated and advanced-stage tumors. Functioning NENs, such as glucagonomas and ACTH-secreting tumors, exhibited a higher thrombotic risk due to their systemic effects on coagulation and inflammation. While deep vein thrombosis, pulmonary embolism, and portal vein thrombosis were the most frequently observed events, rare cases of disseminated intravascular coagulation were also documented. Early recognition and risk stratification, along with targeted prophylactic and therapeutic interventions, are essential to improve patient outcomes. Further research is warranted to elucidate the underlying mechanisms of VTE development in NENs and optimize anticoagulation strategies tailored to this unique patient population.

## 1. Introduction

Neuroendocrine neoplasms (NENs) are a heterogeneous group of tumors, which originate from the neuroendocrine cells dispersed in various organs [1,2]. These tumors vary widely in their biological behavior, from indolent well-differentiated neuroendocrine tumors (NETs) to aggressive, poorly differentiated neuroendocrine carcinomas (NECs), each carrying distinct prognostic and therapeutic implications [3,4].

Venous thromboembolic events (VTEs) represent a significant clinical challenge in cancer care, contributing substantially to morbidity and mortality. Although thromboembolism has been extensively studied in common cancers such as lung, breast, and colorectal malignancies [5,6], its occurrence and impact in patients with NENs remain poorly understood [7,8]. This gap persists despite the rising incidence of NENs, driven by advancements in diagnostic techniques and greater awareness [9]. The unique biological characteristics of NENs, including their ability to secrete bioactive substances, further influence coagulation pathways [10].

The pathogenesis of the prothrombotic state in cancer is multifactorial, involving tumor production of procoagulants and inflammatory cytokines, interaction between tumor cells and blood and endothelial cells, and host responses to the tumor such as inflammation and angiogenesis [5,6,11,12]. In NENs, the mechanisms underlying VTEs are likely similarly complex, yet they remain underexplored [10].

Pancreatic NENs (pan-NENs), in particular, exhibit a high thrombotic burden, often presenting with portal vein thrombosis or other visceral venous thrombi [7,13,14]. Functioning NENs, which secrete bioactive substances and cause systemic effects and specific clinical syndromes, such as glucagonomas and ACTH-secreting tumors, further increase the risk of thrombosis due to associated metabolic disturbances and chronic inflammation [15,16,17]. Moreover, tumor differentiation, functionality, and stage also appear to influence thrombotic risk. Poorly differentiated NECs and moderately to highly differentiated NETs (e.g., G2 and G3) in advanced stages are more frequently associated with thromboembolic events [7]. Despite the potential severity of VTEs in NEN patients, data on the occurrence of thromboembolic events in NENs and current recommendations for antithrombotic prophylaxis are extrapolated from other cancer types, underscoring the urgent need for robust, NEN-specific research to guide risk assessment and management strategies [10].

Given the substantial clinical impact of VTEs in NEN patients, this systematic review consolidates existing evidence on the prevalence, types, and clinical outcomes of VTEs in NEN patients. By examining thrombotic patterns and their association with specific tumor characteristics, this study aims to provide a comprehensive evaluation of thromboembolic events in NENs and lay the groundwork for improving clinical management and outcomes.

## 2. Methods

### 2.1. Search Strategy

A comprehensive systematic review was conducted to evaluate the occurrence of thromboembolism in patients with neuroendocrine neoplasms (NENs). The review followed the PRISMA (Preferred Reporting Items for Systematic Reviews and Meta-Analyses) guidelines to ensure a rigorous and transparent methodology (PROSPERO ID 628388).

### 2.2. Databases and Search Terms

The following databases were searched from inception to the present date: PubMed, Scopus, and Embase. The search strategy was designed to capture all relevant studies related to thromboembolism in neuroendocrine tumors. The search terms included combinations of keywords such as “neuroendocrine neoplasms”, “neuroendocrine tumors”, “neuroendocrine carcinoma”, “NEN”, “NET”, and “thromboembolism”, “thrombosis”, “deep vein thrombosis”, “pulmonary embolism”, and “portal vein thrombosis”. Boolean operators (“AND” and “OR”) were used to capture all relevant studies, including variations in terminology (see Appendix A for the full search strategy).

### 2.3. Inclusion and Exclusion Criteria

This systematic review comprised studies including patients diagnosed with NENs and reporting on the occurrence of venous thromboembolism, defined as deep vein thrombosis (DVT), pulmonary embolism (PE), portal vein thrombosis (PVT), visceral vein thrombosis (VVT), and disseminated intravascular coagulation (DIC). Eligible study types encompassed case reports, case series, cohort studies, retrospective studies, and randomized controlled trials.

Studies were excluded if they did not involve human subjects, were not available in English, or were reviews, editorials, or commentaries without original data. An additional exclusion criterion was the lack of specific data on thromboembolism in NETs.

### 2.4. Data Extraction and Quality Assessment

Two independent reviewers (S.M. and L.G.) screened the titles and abstracts of all identified studies for eligibility. Full-text articles of potentially relevant studies were retrieved and assessed for inclusion. Discrepancies were resolved through discussion and consensus, or by consulting a third reviewer (N.F.).

Data extracted from each study included the study design (e.g., case report, case series, cohort study, retrospective study), characteristics of the NEN (primary site, functionality, differentiation grade), the type and frequency of thromboembolic events, and patient demographics and clinical outcomes.

The quality of the included studies was assessed using standardized tools appropriate for each study design, such as the Newcastle–Ottawa Scale for cohort and case–control studies [18] and the Joanna Briggs Institute checklist for case reports and case series [19].

## 3. Data Synthesis and Analysis

Data were synthesized narratively, and where appropriate, quantitative data were analyzed using a meta-analytic approach, with pooled prevalence and 95% confidence intervals (CIs) calculated using a weighted average. The Wilson method was employed to estimate CIs for individual studies. Studies heterogeneity was evaluated using the I^2^ statistic, and a random-effect model was used if significant heterogeneity was detected. Subgroup analyses were performed based on tumor differentiation grade, functionality, and primary site to explore potential sources of heterogeneity.

## 4. Results

### 4.1. Study Selection

A total of 132 records were identified through database searches, including 73 from PubMed, 32 from Scopus, and 22 from Embase. After removing duplicates, 110 unique records were screened based on their titles and abstracts, resulting in the exclusion of 60 records: 45 were not relevant to VTE in neuroendocrine tumors, 5 were non-human studies, and 10 were reviews, editorials, or commentaries without reporting original data.

The remaining 50 full-text articles were sought for retrieval and assessed for eligibility. Among these, 16 articles were excluded: 10 did not focus on thromboembolism in neuroendocrine tumors, 2 provided insufficient data on thromboembolism, and 5 were case studies that did not meet the inclusion criteria. Ultimately, 33 studies met the inclusion criteria and were included in the systematic review, comprising 26 case reports [20,21,22,23,24,25,26,27,28,29,30,31,32,33,34,35,36,37,38,39,40,41,42,43,44,45], 2 case series [46,47], and 5 retrospective cohort studies [7,13,14,48,49]. The PRISMA flow diagram summarizes the study selection process (Figure 1).

### 4.2. Quality Assessment

The quality of the included studies varied widely. Retrospective cohort studies generally demonstrated higher quality, as assessed using the Newcastle–Ottawa Scale. These studies [7,13,14,48,49] provided more robust data, often featuring well-defined populations, clear outcomes, and good control of confounding factors. In contrast, case reports and case series, assessed using the Joanna Briggs Institute checklist, often lacked methodological rigor due to their small sample sizes, lack of control groups, and potential biases. Scores for case reports and series ranged from 3 to 6 out of 10, reflecting limited generalizability (Table 1).

### 4.3. Study Characteristics

The included studies comprised 26 case reports [20,21,22,23,24,25,26,27,28,29,30,31,32,33,34,35,36,37,38,39,40,41,42,43,44,45], 2 case series [46,47], and 5 retrospective cohort studies [7,13,14,48,49].

### 4.4. Case Reports

The 26 case reports highlighted diverse thrombotic manifestations in NEN patients, emphasizing the heterogeneity of tumor characteristics, primary sites, functionality, and differentiation grades (Table 2).

The pancreas emerged as the most frequent site of NENs associated with thrombosis, accounting for the majority of the reported cases (17 cases) [20,21,22,23,25,27,29,30,33,34,35,36,39,40,41,42]. Other sites included lung NENs (three cases) [24,28,45], of which one was associated with Cushing’s syndrome and two with non-functioning neoplasms [28,45]; two cases were small bowel NETs with carcinoid syndrome [31,43], both with carcinoid heart disease; one case involved colonic NENs [37] and one case was represented by a non-functioning duodenal tumor [26]; one case was a gastric NET [38] and one other was a rectal NET [32].

Both functioning NENs—such as glucagonomas [29,36,41], ACTH-secreting tumors [24,27], serotonin-secreting carcinoids [31,43], calcitonin-secreting neoplasms [39], and VIP- and/or dopamine-secreting tumors [23]—and non-functioning NENs have been implicated in thrombotic events.

Most reported cases involved well-differentiated NENs [20,22,23,24,25,26,29,30,33,34,36,38,39,40,41,43], typically associated with loco-regional thrombosis such as PVT or splenic vein thrombosis. Poorly differentiated NECs demonstrated a more aggressive thrombotic profile, including systemic complications such as DIC and PE [27,37].

The spectrum of all case reports included in this review highlights the variability in tumor characteristics and thrombotic manifestations. Spleno-mesenteric–portal vein thrombosis was the most common thrombotic manifestation [20,21,22,25,30,33,34,35,38,39,40,42], particularly in pancreatic NETs. This was often attributed to tumor invasion into the vascular system, creating a hypercoagulable environment. Pulmonary embolism was the second most frequent type [23,24,27,43,45], frequently associated with lung tumors [24,45] and functioning tumors such as NETs secreting ACTH. Less common manifestations included other visceral vein thrombosis, such as inferior vena cava thrombus extending into the right atrium, Budd–Chiari syndrome, and cerebral sinus venous thrombosis [26,28,29,32,36,37]. Rare presentations included bioprosthetic valve thrombosis, associated with serotonin-secreting NETs [31], and DIC with or without Trousseau’s syndrome [27,37].

### 4.5. Case Series and Retrospective Cohort Studies

Also in the case series and cohort studies, the pancreas remained the predominant site of primary NENs associated with VTE [13,14,47,48,49]. Other sites [7] and cases of multiple endocrine neoplasia type 1 (MEN1) were less frequent [46] but still highlighted a unique thrombotic risk. Both functioning and non-functioning NENs were included, with functioning tumors (e.g., insulinomas, glucagonomas) reported to be linked to increased thrombotic risk due to their systemic effects on metabolic and inflammatory pathways [7,13,14,46].

The studies spanned a spectrum from well-differentiated low-grade tumors (NET G1, G2) to poorly differentiated high-grade NECs, with higher-grade tumors often associated with more severe and systemic thrombotic events.

Common VTE manifestations also included DVT, PE, and PVT across these studies (Table 3).

### 4.6. Prevalence of Venous Thromboembolism in Neuroendocrine Neoplasms

The reported prevalence of VTE in NENs varied across studies. All the studies consistently showed that NEN patients exhibit a significantly higher occurrence of VTE, although no studies comparing this with the general population were available.

The reported frequency of thromboembolism in NENs varies between 5% [14] and 33% [47] across the included retrospective studies. The study by Gervaso et al. [13] reported a 14% incidence of thromboembolism among patients with pancreatic NETs, including DVT, VVT, and PE [13]. Lee et al. found a 12.9% incidence of VTE in patients with various NENs, with DVT occurring in 30.1%, PE in 33.3%, and both in 36.1% of cases [46]. Massironi et al. found a 7.5% prevalence of VTE, with DVT present in 75% of cases and PE in 25%, emphasizing the variability of thrombotic risk across different NEN subtypes [7], with higher thrombotic risk noted in pan-NEN, in advanced tumor stages, and those with moderately to poorly differentiated tumors. Moyana et al. reported a broader prevalence of 13.1% in their series of 61 patients with pan-NENs [49]. Balachandran et al. reported a higher prevalence of 33% of cases presenting visceral vein thrombosis in patients with non-functioning pancreatic NENs [47]. However, this paper focused exclusively on locally advanced or metastatic pan-NENs with vascular invasion, specifically reporting tumor thrombi rather than general VTE. This made its findings less comparable to other studies that included a broader spectrum of NENs presenting thrombotic events during their clinical course.

Therefore, excluding Balachandran et al. [47], a meta-analysis of thrombosis prevalence across six remaining studies [7,13,14,46,48,49] yielded a significant thrombotic risk in patients with NENs. Across the studies, a total of 120 cases of thrombosis were identified among 1092 patients, yielding a pooled prevalence of 10.7% (95% CI: 8.3–13.5%), with only a moderate heterogeneity among the studies (I^2^ = 46.66%), not statistically significant (*p* = 0.09) (Figure 2). The risk is notably pronounced in patients with pan-NENs but is also described for other gastro-entero-pancreatic (GEP) NENs, as well as for lung NENs [10,13,24,28]. It presents in both functioning and non-functioning tumors. Functioning and non-functioning tumors contributed to the thrombotic risk, though specific hormonal syndromes such as those caused by glucagonoma and serotonin-secreting tumors appeared to exacerbate the prothrombotic state [7,14,29,36,49].

Lastly, the prevalence of VTE varies according to tumor grade, with moderate to poorly differentiated NENs associated with a greater thrombotic burden, often exhibiting a higher risk of systemic and aggressive thrombotic complications.

### 4.7. Types of Thromboembolic Events

VTE manifestations in NEN patients were diverse. PVT was the most common, particularly in pan-NENs, attributed to tumor invasion into the vascular system. PE and DVT were frequently associated with functional tumors or advanced-stage disease. Rare presentations included cerebral venous sinus thrombosis, bioprosthetic valve thrombosis, and thrombosis in the internal jugular and subclavian veins.

### 4.8. Clinical Outcomes

Across the different studies, patient outcomes varied widely depending on the type and severity of thromboembolism, as well as tumor characteristics.

Localized thrombi, such as those reported by Nguyen et al. [22] and Prakash et al. [48], particularly in well-differentiated NETs, often allow for more favorable outcomes with appropriate management. For example, PVT associated with pan-NETs was effectively managed with anticoagulation therapy or surgical resection in the cases reported [22,48].

Several case reports highlighted severe complications such as multiple organ infections, DIC, and death. For example, Yoshihara et al. [27] described a case of ACTH-secreting pancreatic neuroendocrine carcinoma (NEC) with multiple organ infections, DVT, PE, and DIC, resulting in poor prognosis. In another report, Ohmura et al. described a case of NEC of the colon in a 69-year-old patient, in which disease progression led to DIC associated with Trousseau’s syndrome, culminating in multiple small infarcts in the cerebral cortex and white matter [37]. Additionally, DIC has been documented in another case report by The et al. of a pancreatic NEN, where DIC initially developed following tumor biopsy and subsequently reoccurred after the initiation of carboplatin and etoposide chemotherapy [44]. During the second recurrence, DIC resulted in severe bleeding, which was fatal despite intensive treatment, and the malignancy was considered the underlying cause of DIC [44].

Hormonal activity in functioning tumors, such as serotonin-secreting carcinoids, exacerbated the risk of complications like carcinoid heart disease and prosthetic valve thrombosis. For instance, a complication of serotonin-secreting carcinoids is represented by carcinoid heart disease (CHD) which markedly increases the risk of thrombotic events [31,43]. CHD’s serotonin-induced fibrosis of right-sided heart valves leads to significant valvular dysfunction. Elevated serotonin levels associated with carcinoid syndrome promote platelet aggregation and clot formation, creating a hypercoagulable state that exacerbates thrombotic risk [31]. Patients with CHD and bioprosthetic valves are particularly susceptible to prosthetic valve thrombosis, underscoring the importance of meticulous anticoagulation management [31]. Again, Mohd Nasri et al. described pulmonary embolism in a patient with colonic NET-induced CHD [43]. As already highlighted in the various case reports, poorly differentiated NECs are associated with more aggressive thrombotic profiles and systemic complications, leading to poorer prognoses [27,37,44,45]. In contrast, well-differentiated NETs, although still associated with a prothrombotic state, tend to involve more localized events with a lower risk of life-threatening sequelae. Indeed, for localized thrombi, survival rates improve significantly when managed promptly and appropriately.

### 4.9. Possible Pathophysiological Mechanisms

The pathophysiology of thrombosis in NENs is multifaceted, reflecting interactions between tumor biology, host responses, and the unique features of NENs [49]. These tumors often secrete bioactive substances, including hormones and cytokines, that disrupt coagulation pathways, foster inflammation, and promote thrombosis.

Functioning tumors, such as glucagonomas and serotonin-secreting tumors, exacerbate the thrombotic risk through systemic effects. For instance, elevated glucagon levels in glucagonoma syndrome disrupt protein and lipid metabolism, inducing a hypercoagulable state, while deficiencies in amino acids and zinc further impair coagulation homeostasis [17]. Similarly, serotonin-secreting tumors promote fibrosis and vascular obstruction, while ACTH-secreting tumors increase clotting factors and impair fibrinolysis, further tipping the balance toward thrombosis [16,24,27].

Endothelial dysfunction is a central driver of thrombosis in NENs. Tumor cells damage endothelial cells, increasing adhesion molecule expression and releasing inflammatory cytokines such as interleukin-6 (IL-6) and tumor necrosis factor-alpha (TNF-α), which amplify coagulation and inhibit fibrinolysis [51]. Poorly differentiated NECs show high expression of tissue factor (TF), a critical initiator of coagulation, further increasing thrombotic potential [10].

Additionally, the genetic landscape of NENs significantly influences thrombogenic potential. Examples include mutations in tumor suppressor genes, such as p53, and alterations in pathways regulating angiogenesis and hypoxia and promoting TF expression and procoagulant activity [52,53]. The genetic heterogeneity seen in pheochromocytomas, paragangliomas, and other NENs implicates diverse pathways, including angiogenesis, metabolism, and hypoxia signaling, in their prothrombotic tendencies [54].

Moreover, the tumor microenvironment plays a key role. The interaction between tumor cells and the extracellular matrix (ECM) contributes to thrombogenesis. Components of the ECM, such as collagen and fibronectin, bind coagulation factors and platelets, facilitating clot formation. Dysregulated proteases and inhibitors in the tumor microenvironment disrupt fibrinolysis, further enhancing thrombotic risk [55,56]. Notably, small bowel NENs (sbNENs) often induce fibrosis, locally and distantly. This fibrotic process, driven by serotonin, growth factors, and peptides, promotes fibroblast activation and matrix remodeling, creating a thrombogenic niche [57,58,59].

Notably, NENs are also characterized by high vascularization, partly mediated by overexpression of proangiogenic molecules such as VEGF, IL-6, and TGF-β. These factors stimulate TF expression, driving thrombin generation and clot formation. Additionally, microparticles (MPs) released by NEN cells, enriched with TF and other procoagulants, interact with platelets and endothelial cells, amplifying thrombosis risk [60]. MPs are small vesicles released by various cells, including NENs, containing procoagulant and anticoagulant molecules. NEN-derived MPs are enriched with TF and other procoagulant factors, promoting thrombin generation. MPs can interact with platelets and endothelial cells, further contributing to thrombosis. Elevated levels of fibrinogen, decreased activity of natural anticoagulants like protein C and antithrombin, and impaired fibrinolysis (clot breakdown) contribute to the prothrombotic state in NENs.

Platelets play a critical role in tumor angiogenesis, which closely correlates with thrombogenesis. Tumor–platelet interactions enhance growth, metastasis, and clot formation. Targeting platelet function may offer therapeutic opportunities, including antiplatelet agents with antiangiogenic effects [60]. The interaction between tumor cells, platelets, and the coagulation system can enhance tumor growth, metastasis, and angiogenesis. Although not specific to NENs, these findings suggest broader implications for thrombosis management in cancer patients.

Therefore, the pathophysiology of thrombosis in NENs is multifactorial, involving direct and indirect effects of the tumor on the coagulation system. Understanding these mechanisms is crucial for developing targeted therapies that address the underlying causes of thrombosis in NEN patients, potentially improving their clinical outcomes and quality of life.

## 5. Discussion

This systematic review and meta-analysis highlights the significant thrombotic burden associated with NENs, emphasizing the intricate interplay between tumor biology, systemic coagulation mechanisms, and patient outcomes. The meta-analysis of the prevalence of VTE was estimated at 11.1% (95% CI: 9.07–13.53%), confirming that VTEs are a frequent and clinically relevant complication in NEN patients.

The pathogenesis of thrombosis in NENs is multifaceted, involving tumor-secreted factors, inflammation, and direct vascular invasion. Functioning tumors, such as serotonin-secreting carcinoid tumors [7,31,43] and glucagonomas [7,29,36,41], may exacerbate thrombotic risk through metabolic disturbances, endothelial activation, and coagulopathy. These mechanisms, combined with metabolic disturbances, create a hypercoagulable environment that predisposes patients to thromboembolic events such as DVT, PE, and PVT. Poorly differentiated NECs further complicate this picture, as their aggressive nature is often linked to severe thrombotic complications, including DIC, which carries a poor prognosis [27,37,44].

Certain predisposing factors such as liver cirrhosis, pregnancy, myeloproliferative neoplasms, Factor V Leiden mutation, protein C or protein S deficiency, and the JAK2V617F mutation may coexist with the NEN in some NEN patients, further increasing susceptibility to thrombosis. For instance, a case report described a 25-year-old patient with a non-functioning grade 2 pancreatic NET and a JAK2V617F mutation, presenting with splenic vein thrombosis at diagnosis [61]. Such examples highlight the need to consider additional risk factors when managing thrombosis in NEN patients.

While NENs carry a substantial risk of TEs, this appears to be lower than in high-risk cancers such as pancreatic adenocarcinoma or gastric malignancies, where thrombotic rates range from 15 to 41% [62]. The lower overall prevalence may partly reflect the indolent nature of well-differentiated NENs, which constitute the majority of cases [13]. However, NECs, with their more aggressive biology, demonstrate thrombotic risks comparable to those seen in other high-grade malignancies [13,52]. On the other hand, it is important to note that none of the included studies directly compared the incidence of VTE in patients with NENs to that in the general population. Notably, Cronin-Fenton et al. reported a VTE incidence rate of 4.7 per 1000 person-years in the general population [63], highlighting that patients with NENs likely face an elevated thrombotic risk, although the exact magnitude of this increase remains to be quantified through comparative studies. Future research should incorporate control groups to better quantify the thrombotic risk associated with NENs and identify tumor-specific mechanisms driving VTE development.

Our findings emphasize the need for increased clinical vigilance in identifying and managing VTEs in NEN patients, as well as a greater awareness of the complication in NEN physicians. Thrombosis in this population is often under-recognized, despite its significant contribution to morbidity and mortality. Early detection and prompt intervention, including anticoagulation therapy, are crucial for improving outcomes. Risk stratification based on tumor site, functionality, and grade is also essential for guiding management decisions. For instance, pancreatic NENs, particularly those at advanced stages or with poorly differentiated histology, are consistently associated with a higher thrombotic burden. Moreover, the role of new cancer therapies in promoting thrombotic risk deserves special attention. Targeted treatments, such as VEGF inhibitors, immunotherapies, and cytotoxic agents, may further alter the hemostatic balance through mechanisms such as endothelial damage, cytokine release, and platelet activation [64,65]. In addition, the use of indwelling catheters, which are often required for prolonged treatment regimens, increases the likelihood of catheter-associated thrombosis [66]. These factors highlight the need for vigilant monitoring and individualized anticoagulation strategies to mitigate the risk of venous thromboembolism in patients with NENs receiving modern therapies.

The management of VTEs in NEN patients presents unique challenges. Anticoagulation therapy, while effective in preventing and treating thromboembolic events, poses a significant bleeding risk, particularly in patients with advanced or metastatic disease. In addition, just in these patients, the frequent need for intramuscular injections of somatostatin analogs further complicates anticoagulation management.

Surgical interventions, including resection of tumor-associated thrombi, can be curative in localized cases but are not without risk. Moreover, the decision to initiate prophylactic anticoagulation must balance the thrombotic risk against the potential for hemorrhage, particularly in patients with functional tumors that may exacerbate coagulopathy.

This review has several strengths, including the systematic evaluation of thromboembolic events across diverse NEN subtypes and the inclusion of both case reports and cohort studies to provide a comprehensive perspective. However, some limitations should be acknowledged. Firstly, the heterogeneity of the included studies, particularly in terms of tumor types, functionality, and differentiation grade, poses challenges in generalizing the findings. Secondly, the absence of control groups in most studies limits direct comparisons between NEN patients and the general population or other cancer types. Lastly, the retrospective nature of many studies and the reliance on small case series limit the robustness of the prevalence estimates.

In conclusion, the findings of this review highlight the need for prospective studies investigating the incidence and risk factors for VTEs in NEN patients compared with both the general population and other malignancies. Such studies should incorporate standardized criteria for diagnosing and classifying thrombosis, as well as control groups to better contextualize the observed prevalence. Future research should focus on prospective studies to quantify the true burden of thrombosis in NEN patients and identify high-risk populations. Comparative analyses between NENs and other malignancies may elucidate tumor-specific mechanisms driving thrombosis. Therefore, studies evaluating the efficacy and safety of thromboprophylaxis and of novel anticoagulant therapies in this unique population are needed, particularly in patients with advanced-stage disease or poorly differentiated NECs.

## 6. Conclusions

This review highlights the significant burden of thrombosis in NEN patients, particularly those with advanced-stage or poorly differentiated tumors. The findings underscore the need for heightened clinical vigilance, proactive screening, and tailored management strategies to mitigate thrombotic risk in this population.

## Figures and Tables

**Figure 1 cancers-17-00212-f001:**
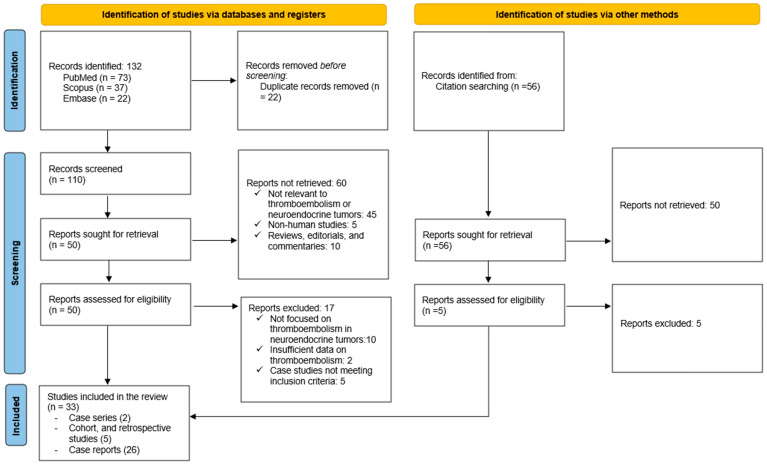
Prisma flow-chart.

**Figure 2 cancers-17-00212-f002:**
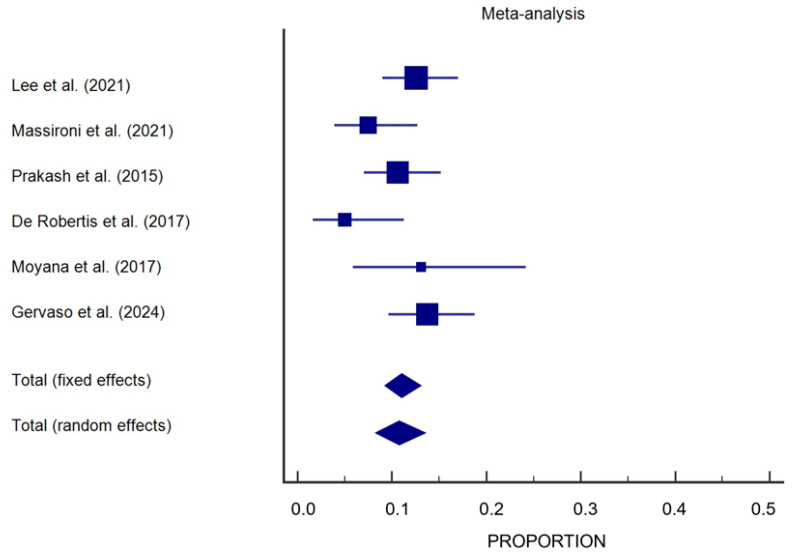
Thrombosis prevalence in NEN studies (squared symbols)The meta-analysis revealed a pooled VTE prevalence (diamond symbol) of 11.1% (95% CI: 9.07–13.53%) [7,13,14,46,48,49].

**Table 1 cancers-17-00212-t001:** Quality assessment of the included studies.

Study	Type of Study	Assessment Tool	Quality Score	Comments
Gervaso et al. (2024) [13]	Retrospective Cohort Study	Newcastle–Ottawa Scale	7/9	High quality, good selection, and comparability.
Massironi et al. (2021) [7]	Retrospective Cohort Study	Newcastle–Ottawa Scale	7/9	High quality, good control of confounding factors.
Moyana et al. (2017) [49]	Retrospective Study	Newcastle–Ottawa Scale	6/9	Moderate quality, some risk of bias due to retrospective design.
De Robertis et al. (2017) [14]	Retrospective Study	Newcastle–Ottawa Scale	7/9	High quality, clear outcome assessment and follow-up.
Prakash et al. (2015) [48]	Retrospective Study	Newcastle–Ottawa Scale	6/9	Moderate quality, potential selection bias.
Balachandran et al. (2009) [47]	Case Series	Joanna Briggs Institute checklist	5/10	Moderate quality, limited generalizability.
Lee et al. (2021) [46]	Case series	Newcastle–Ottawa Scale	8/9	High quality, well-defined population and outcomes.
Busch et al. (2013) [45]	Case Report	Joanna Briggs Institute checklist	4/10	Low quality, single case, lack of control group.
Tian et al. (2021) [20]	Case Report	Joanna Briggs Institute checklist	3/10	Low quality, single patient, limited methodological rigor.
Miyata et al. (2020) [21]	Case Report	Joanna Briggs Institute checklist	4/10	Low quality, limited information on methodology.
Nguyen et al. (2005) [22]	Case Report	Joanna Briggs Institute checklist	3/10	Low quality, lack of detailed patient information.
Nilubol et al. (2016) [23]	Case Report	Joanna Briggs Institute checklist	5/10	Moderate quality, some methodological details provided.
Yang et al. (2019) [24]	Case Report	Joanna Briggs Institute checklist	4/10	Low quality, single case, limited details.
Zhu et al. (2020) [25]	Case Report	Joanna Briggs Institute checklist	4/10	Low-quality, single-case report.
Yu et al. (2018) [26]	Case Report	Joanna Briggs Institute checklist	5/10	Moderate quality, detailed case description.
Yoshihara et al. (2022) [27]	Case Report	Joanna Briggs Institute checklist	4/10	Low quality, lack of control, and generalizability.
Liaqat et al. (2021) [28]	Case Report	Joanna Briggs Institute checklist	3/10	Low-quality, single-case report.
Delli Colli et al. (2022) [29]	Case Report	Joanna Briggs Institute checklist	5/10	Moderate quality, detailed clinical course.
Rodriguez et al. (2014) [30]	Case Report	Joanna Briggs Institute checklist	4/10	Low quality, single case, limited methodological rigor.
Hollander et al. (2019) [31]	Case Report	Joanna Briggs Institute checklist	5/10	Moderate quality, detailed patient history, and outcome.
Hurtado-Cordovi et al. (2013) [32]	Case Report	Joanna Briggs Institute checklist	4/10	Low quality, limited generalizability.
Bedirli et al. (2004) [33]	Case Report	Joanna Briggs Institute checklist	3/10	Low quality, limited methodological details.
Barbier et al. (2010) [34]	Case Report	Joanna Briggs Institute checklist	4/10	Low quality, single patient, lack of control.
Gao et al. (2021) [35]	Case Report	Joanna Briggs Institute checklist	5/10	Moderate quality, detailed clinical and imaging findings.
Han et al. (2016) [36]	Case Report	Joanna Briggs Institute checklist	5/10	Moderate quality, comprehensive clinical presentation and follow-up.
Ohmura et al. (2023) [37]	Case Report	Joanna Briggs Institute checklist	4/10	Low quality, single case, limited detail on long-term outcomes.
Sninate et al. (2021) [38]	Case Report	Joanna Briggs Institute checklist	4/10	Low quality, single case, detailed imaging findings but limited generalizability.
Piłka et al. (2024) [39]	Case Report	Joanna Briggs Institute checklist	5/10	Moderate quality, robust discussion of clinical challenges and therapeutic considerations.
Garcia Soria et al. (2023) [40]	Case Report	Joanna Briggs Institute checklist	4/10	Low quality, brief case description, limited details.
Eldor et al. (2011) [41]	Case Report	Joanna Briggs Institute checklist	6/10	Moderate quality, cumulative experience of multiple cases over years.
Kumar et al. (2021) [42]	Case Report	Joanna Briggs Institute checklist	5/10	Moderate quality, detailed imaging and clinical findings.
Mohd Nasri et al. (2024) [43]	Case Report	Joanna Briggs Institute checklist	4/10	Low-quality, single-case, brief clinical presentation.
Teh et al. (2012) [44]	Case Report	Joanna Briggs Institute checklist	5/10	Moderate quality, robust description of the case and therapeutic outcomes.

**Table 2 cancers-17-00212-t002:** Case reports on venous thromboembolic events (VTEs), including pulmonary embolism (PE), deep vein thrombosis (DVT), and visceral vein thrombosis (VVT) in patients with neuroendocrine neoplasm (NEN).

Reference	Primary Neuroendocrine Tumor	Functioning	Differentiation Grade	Type of Thrombosis
Tian et al. (2021) [20]	Pancreatic NET	Non-functioning	Well-differentiated	Portal vein thrombosis
Miyata et al. (2020) [21]	Pancreatic NET	Non-functioning	N/A	Splenic vein thrombosis
Nguyen et al. (2005) [22]	Pancreatic NET	Non-functioning	Well-differentiated	Visceral vein thrombosis (portal vein)
Nilubol et al. (2016) [23]	Pancreatic VIPoma	Functioning (VIP, Dopamine)	Well-differentiated	Pulmonary embolism
Yang et al. (2019) [24]	ACTH-secreting bronchial carcinoid	Functioning (ACTH)	Well-differentiated	Pulmonary embolism
Zhu et al. (2020) [25]	Pancreatic NET	Non-functioning	Well-differentiated	Extensive tumor thrombosis in the portal vein
Yu et al. (2018) [26]	Duodenal NET	Non-functioning	Well-differentiated	IVC neoplastic thrombosis (with right atrium)
Yoshihara et al. (2022) [27]	Pancreatic NEC	Functioning (ACTH)	Poorly differentiated	DVT, PE, DIC
Liaqat et al. (2021) [28]	Pulmonary NEC	Non-functioning	Poorly differentiated	Internal jugular and subclavian vein thrombosis
Delli Colli et al. (2022) [29]	Pancreatic Glucagonoma	Functioning (Glucagon)	Well-differentiated	Cerebral sinus venous thrombosis
Rodriguez et al. (2014) [30]	Pancreatic NET	Non-functioning	Well-differentiated	Splenic vein thrombosis
Hollander et al. (2019) [31]	Small intestine NET (Carcinoid heart disease)	Functioning (Serotonin)	N/A	Bioprosthetic valve thrombosis
Hurtado-Cordovi et al. (2013) [32]	Rectal NET	N/A	N/A	Budd–Chiari syndrome
Bedirli et al. (2004) [33]	Pancreatic NET	Non-functioning	Well-differentiated	Portal vein thrombosis
Barbier et al. (2010) [34]	Pancreatic NET	Non-functioning	Well-differentiated G2	Spleno-mesenteric–portal thrombosis
Gao et al. (2021) [35]	Pancreatic NEC	N/A	G3	Portal vein thrombosis
Han et al. (2016) [36]	Pancreatic Glucagonoma	Functioning (Glucagon)	Well-differentiated G2	IVC thrombosis
Ohmura et al. (2023) [37]	Colonic NEC	N/A	Poorly differentiated	DIC and cerebral thrombosis in Trousseau’s syndrome
Sninate et al. (2021) [38]	Stomach NET	N/A	Well-differentiated G1	Splenic vein thrombosis
Piłka et al. (2024) [39]	Pancreatic NET	Functioning (Calcitonin)	Well-differentiated G2	Portal and splenic vein thrombosis
García Soria et al. (2023) [40]	Pancreatic NET	N/A	Well-differentiated	Splenic vein thrombosis
Eldor et al. (2011) [41]	Pancreatic Glucagonoma	Functioning (Glucagon)	Well-differentiated G2	DVT
Kumar et al. (2021) [42]	Pancreatic NEC	Non-functioning	G3	Splenic vein thrombosis
Mohd Nasri et al. (2024) [43]	Colonic NET (Carcinoid heart disease)	Functioning (Serotonin)	Well-differentiated G2	Pulmonary embolism
Teh et al. (2012) [44]	Possible pancreatic NEC	Non-functioning	Poorly differentiated	DIC
Busch et al. (2013) [45]	Lung	Non-functioning	Poorly differentiated	Progressive arterial embolisms

NET = neuroendocrine tumor; NEC = neuroendocrine carcinoma; DIC = disseminated intravascular coagulation; IVC = inferior vena cava.

**Table 3 cancers-17-00212-t003:** Case series, cohort studies, and retrospective studies on thromboembolic events (VTEs), including pulmonary embolism (PE), deep vein thrombosis (DVT), and visceral vein thrombosis (VVT) in patients with neuroendocrine neoplasm (NEN).

Type of Study	Reference	Primary NENs	Functioning	Differentiation Grade	Frequency of Thrombosis (%)	Type of Thrombosis
Case Series						
	Lee et al. (2021) [46]	Various NENs in MEN-1	Functioning and Non-functioning	Well-differentiated and Poorly differentiated	12.9	DVT
	Balachandran et al. (2012) [47]	Pancreatic NENs	Non-functioning	N/A	33	VVT
Retrospective Cohort Studies						
	Gervaso et al. (2024) [13]	Pancreatic NENs	Functioning and Non-functioning	Well-differentiated and Poorly differentiated	14	DVT, PE, and VVT
	Prakash et al. (2015) [48]	Pancreatic NET	Non-functioning	44.4% G2, 11.1% G3	11 (26/245)	Portal venous thrombosis
	Massironi et al. (2021) [7]	GEP-NENs (Gut, Pancreas)	Functioning and Non-functioning	25% G1, 50% G2, 16.7% G3	7.5	DVT and PE
	DeRobertis et al. (2018) [50]	Pancreatic NENs	Functioning and Non- functioning	66.7% G2, 33.3% G3	5	DVT
	Moyana et al. (2017) [49]	Pancreatic NENs	Non- functioning	87.5% G2, 12.5% G3	13.1 (8/61)	Splenic vein thrombosis

GEP = gastro-entero-pancreatic; NEN = neuroendocrine neoplasm; MEN-1 = multiple endocrine neoplasia type 1.

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
