# Peer review of "Venous Thromboembolism in Patients with Neuroendocrine Neoplasms: A Systematic Review of Incidence, Types, and Clinical Outcomes"

_cancers, 2025, doi:10.3390/cancers17020212_

Round 1

Reviewer 1 Report

Comments and Suggestions for Authors

This comprehensive systematic review and meta-analysis evaluates the incidence, types, and clinical outcomes of venous thromboembolic events (VTEs) in patients with neuroendocrine neoplasms (NENs). NENs are a heterogeneous group of tumors with unique biological characteristics and complications, including thromboembolism. A systematic search of PubMed, Scopus, and Embase was conducted to identify studies on TEs in NENs. Eligible studies included case reports, case series, and retrospective cohort studies reporting VTEs, including deep vein thrombosis (DVT), pulmonary embolism (PE), and visceral vein thrombosis (VVT). Data were extracted on tumor site, functionality, differentiation grade, and VTE type. Thirty-tree studies were included, comprising 26 case reports, 2 case series, and 5 retrospective cohort studies. VTE prevalence ranged from 7.5% to 33% across studies. The most common VTEs were DVT, PE, and portal vein thrombosis (PVT). A meta-analysis revealed a pooled VTE prevalence of 11.1% (95% CI: 9.07–13.53%). Pancreatic NENs exhibited the highest thrombotic burden, particularly in poorly differentiated and advanced-stage tumors. Besides, functioning tumors, including glucagonomas and ACTH-secreting NENs, were strongly associated with VTEs.

The topic of the article is completely covered. The meta-analysis confirmed that VTEs are a frequent and clinically relevant complication in NEN patients. In addition the authors discuss in detail the multifactorial pathogenesis  and management of VTEs in patients with NETs.

 All 60 references are relevant, there are 4 self-citations and 29 are recent publications (within the last 5 years). Tables 1, 2, 3, Figures 1 and 2 and Appendix are appropriate - they properly show the data and are easy to understand.

The paper is clear, comprehensive, relevant to the field and well structured. The conclusions are consistent with the evidence and arguments presented. The paper is worth publishing.

 There is only one mistake:

18. Wells GA SB, O'Connell D. The Newcastle-O􀄴awa Scale (NOS) for Assessing the Quality of Nonrandomised Studies in Meta- 443

analyses. O􀄴awa: O􀄴awa Hospital Research Institute. p. Available at 444

h􀄴p://www.ohri.ca/programs/clinical_epidemiology/oxford.asp. 445

19. Institute. JB. Checklist for Case Reports. 2017.

Author Response

Thank you very much for your thorough and positive evaluation of our manuscript. We appreciate your thoughtful comments and constructive suggestions, which have helped us further refine and improve the paper. Please find our detailed responses below, along with the corresponding revisions highlighted in the re-submitted manuscript.

Comment 1:
"The paper is clear, comprehensive, relevant to the field, and well-structured. The conclusions are consistent with the evidence and arguments presented. The paper is worth publishing. There is only one mistake:

  1. Wells GA SB, O'Connell D. The Newcastle-Ottawa Scale (NOS) for Assessing the Quality of Nonrandomised Studies in Meta-analyses. Ottawa: Ottawa Hospital Research Institute. p. Available at h?p://www.ohri.ca/programs/clinical_epidemiology/oxford.asp.

  2. Institute. JB. Checklist for Case Reports. 2017."

Response 1:
Thank you for identifying this citation error. We have corrected the reference formatting as follows:

  1. Wells GA, Shea B, O'Connell D, Peterson J, Welch V, Losos M, Tugwell P. The Newcastle-Ottawa Scale (NOS) for assessing the quality of nonrandomised studies in meta-analyses. Ottawa: Ottawa Hospital Research Institute; 2017. Available from: http://www.ohri.ca/programs/clinical_epidemiology/oxford.asp.

  2. Joanna Briggs Institute. JB Checklist for Case Reports. Adelaide, Australia: Joanna Briggs Institute; 2017.

These corrections can be found in the References section of the revised manuscript.

Reviewer 2 Report

Comments and Suggestions for Authors

The merit of this retrospective review is that it brings together studies dealing with thrombotic complications that are still little explored in a spectrum of rare NEN tumors, The articles were subject to strict selection however,  bringing back retrospective studies and case/series reports of uneven quality. This point combined with the retrospective nature are the limitations of the review, as the authors rightly mentioned in the discussion. The aim is to raise awareness of these complications, for better management in a field that is still poorly understood

Line 50

Worth mentioning

New cancer treatments also play an important role in the multifactorial process, especially when using indwelling catheters

Line 63 and 327-328

Editorial the police of these sentences must be corrected

Figure 1: some problems with the figures 

Identification: Record identified 132 instead of 13245

Screening: Reports excluded: 17 instead of 16

Line 167: functioning tumors consist in… (verb is missing?)

Line 174: to be modified:  The spectrum of all cases reports instead of these cases because these cases seems to the refer to the NEC cases cited in the former sentence 

Line 174: To be changed: The range of all cases reports instead of “these cases” because “these cases” seem to refer to the NEC cases only cited in the previous sentence

Line 322

Antithrombin is no more use “III” must be erased , Antithrombin alone is OK

Line 324-326

Although not specific to NEN,…

Author Response

Thank you very much for your careful review of our manuscript and your valuable comments. We appreciate your constructive feedback, which has allowed us to refine the paper further. Below, we have provided detailed responses to each of your points, along with the corresponding revisions highlighted in the re-submitted manuscript.

Comment 1:
“The merit of this retrospective review is that it brings together studies dealing with thrombotic complications that are still little explored in a spectrum of rare NEN tumors. The articles were subject to strict selection however, bringing back retrospective studies and case/series reports of uneven quality. This point combined with the retrospective nature are the limitations of the review, as the authors rightly mentioned in the discussion. The aim is to raise awareness of these complications, for better management in a field that is still poorly understood.”

Response 1:
We appreciate this positive evaluation of the manuscript and the acknowledgment of its contribution to highlighting thrombotic complications in NEN patients. We agree with the reviewer’s assessment that the retrospective design and variable quality of the included studies represent limitations, and we have emphasized this point in the Discussion section (pag 16, line 420)

Comment 2 (Line 50):
“Worth mentioning: New cancer treatments also play an important role in the multifactorial process, especially when using indwelling catheters.”

Response 2:
We thank the reviewer for this suggestion. We have added a sentence to highlight the role of new cancer treatments and the use of indwelling catheters in the multifactorial process contributing to thrombosis. This revision can be found in the Discussion (page 15, lines 397-404)

Comment 3 (Lines 63 and 327–328):
“Editorial the police of these sentences must be corrected.”

Response 3:
Thank you for pointing this out. We have carefully reviewed and corrected the editorial style of these sentences. 

Comment 4 (Figure 1):
“Some problems with the figures: Identification - Record identified 132 instead of 13245. Screening - Reports excluded: 17 instead of 16.”

Response 4:
We appreciate this observation and have corrected Figure 1 as follows: Identification now reads 132 records identified. Screening now reports 17 exclusions instead of 16.

Comment 5 (Line 167):
“Functioning tumors consist in… (verb is missing?)”

Response 5:
Thank you for highlighting this issue. We appreciate your suggestion, and we have revised the sentence for clarity and grammatical accuracy. The updated text can be found in the Results section (line 182) and now explicitly states that both functioning and non-functioning NENs are implicated in thrombotic events. Furthermore, we clarified the definition of functioning tumors in the Introduction (page 2, line 72) to ensure consistency throughout the manuscript.

Comment 6 (Line 174):
“To be modified: The spectrum of all case reports instead of these cases because these cases seems to refer to the NEC cases cited in the former sentence.”

Response 6:
We have clarified the sentence to avoid confusion. It now reads:
"The spectrum of all case reports included in this review highlights the variability in tumor characteristics and thrombotic manifestations." See the Results section (line 190).

Comment 7 (Line 322):
“Antithrombin is no more use ‘III’ must be erased, Antithrombin alone is OK.”

Response 7:
We appreciate this correction and have removed ‘III’ to refer to Antithrombin appropriately.

Comment 8 (Lines 324–326):
“Although not specific to NEN,….”

Response 8:
We have rephrased the sentence for improved readability and flow. The revised sentence reads:
"Although not specific to NENs, these findings suggest broader implications for thrombosis management in cancer patients."

Reviewer 3 Report

Comments and Suggestions for Authors Apparently, the NET populations are not being compared to control populations. What is the TE rate in normal (non-cancer) populations and patients with non-NET cancers?

Author Response

We would like to sincerely thank the reviewer for the time spent reviewing our manuscript.

Comment:
"Apparently, the NET populations are not being compared to control populations. What is the TE rate in normal (non-cancer) populations and patients with non-NET cancers?"

Response:
Thank you for raising this important point. We agree that none of the studies included in our review directly compared the incidence of VTE in patients with NENs to that observed in the general population or in patients with non-NEN cancers. As noted by Cronin-Fenton et al., the incidence rate of VTE in the general population is approximately 4.7 per 1000 person-years. This provides useful context, highlighting that patients with NENs likely face an elevated thrombotic risk, although the exact magnitude of this increase remains to be formally quantified through comparative studies. We sincerly appreciate the reviewer’s observation and believe this addition strengthens the manuscript by clearly acknowledging this limitation and emphasizing directions for future research.To address this limitation, we have added the following statement in the Discussion section (page 15, lines 382-388).